# Conventional CD4[+] T cells present bacterial antigens to induce cytotoxic and memory CD8[+] T cell responses

Aránzazu Cruz-Adalia[1], Guillermo Ramirez-Santiago[1,2], Jesús Osuna-Pérez[1], Mónica Torres-Torresano[1], Virgina Zorita[3], Ana Martínez-Riaño[4], Viola Boccasavia[4], Aldo Borroto[4], Gloria Martínez del Hoyo[3], José María González-Granado[3,5], Balbino Alarcón[4], Francisco Sánchez-Madrid[6] & Esteban Veiga [1]

Bacterial phagocytosis and antigen cross-presentation to activate CD8[+] T cells are principal functions of professional antigen presenting cells. However, conventional CD4[+] T cells also capture and kill bacteria from infected dendritic cells in a process termed transphagocytosis (also known as transinfection). Here, we show that transphagocytic T cells present bacterial antigens to naive CD8[+] T cells, which proliferate and become cytotoxic in response. CD4[+] T-cell-mediated antigen presentation also occurs in vivo in the course of infection, and induces the generation of central memory CD8[+] T cells with low PD-1 expression. Moreover, transphagocytic CD4[+] T cells induce protective anti-tumour immune responses by priming CD8[+] T cells, highlighting the potential of CD4[+] T cells as a tool for cancer immunotherapy.

[1] Department of Molecular & Cellular Biology, Centro Nacional de Biotecnología; Consejo Superior de Investigaciones Científicas (CNB-CSIC), Darwin 3, 28049 Madrid, Spain. [2] Hospital de Santa Cristina, Instituto de Investigación Sanitaria Princesa, 28009 Madrid, Spain. [3] Centro Nacional de Investigaciones Cardiovasculares Carlos III (CNIC), Melchor Fernández Almagro 3, 28029 Madrid, Spain. [4] Department of Cell Biology and Immunology, Centro de Biología Molecular Severo Ochoa (CBMSO); Nicolás Cabrera 1, Universidad Autónoma de Madrid, 28049 Madrid, Spain. [5] Instituto de Investigación Sanitaria Hospital 12 de Octubre (imas12), 28041 Madrid, Spain. [6] Hospital de la Princesa, Instituto de Investigación Sanitaria Princesa, 28006 Madrid, Spain. Correspondence and requests for materials should be addressed to A.C.-A. (email: acruz@cnb.csic.es) or to E.V. (email: eveiga@cnb.csic.es)

The immune response to intracellular bacterial pathogens (such as *Listeria monocytogenes*) and cancer are similar in many ways, and require the participation of CD8+ effector T cells[1]. Specific T-cell receptors (TCR) on CD8+ T cells recognize antigens exposed in the context of major histocompatibility complex I (MHC-I) on the surface of antigen presenting cells (APC)[2]. As almost all cells express MHC-I, malignant cells (expressing neo-antigens) or infected cells (expressing pathogen antigens) can be eliminated by effector CD8+ T cells. To eliminate these types of cells directly, antigen-specific CD8+ T cells must first be activated (or 'primed') by professional APCs[3]. APC presentation of exogenous antigens, such as bacterial antigens, via MHC-I is termed cross-presentation[3,4], and is crucial for CD8+ T-cell responses to infectious diseases and in cancer. Although many cell types are reported to be able to cross-present antigens, the most efficient APCs[3–5] in vivo are dendritic cells (DC).

Antigen is presented during generation of the immune synapse (IS), formed by the intimate contact of T cells and antigen-loaded APCs, and organized into concentric rings of multimolecular assemblies termed the supramolecular activation cluster (SMAC)[6]. After activation, some CD8+ T cells develop into memory cells, which respond more rapidly and efficiently to later infections[7] and are of major relevance in tumour immunotherapies. The main APCs involved in memory CD8+ T-cell generation are DCs, which can therefore be used in cellular therapies as anti-cancer vaccines[8]. Other immunotherapies that provide survival benefits utilize the checkpoint inhibitors (anti-PD-1, anti-PD-L1, or anti-CTLA-4 antibodies), which block CD8+ T-cell inhibition promoted by the immunosuppressive tumour environment[9]. However, these therapies clinically benefit only 25–30% of patients, and many cancer types are resistant[10]. New strategies are therefore needed to improve antitumour immunotherapy[11].

CD4+ T cells seem to be necessary for the generation of memory CD8+ T cells[12,13], although some debate exists[14]. Interaction of the CD40 ligand (CD40L) on the CD4+ T-cell surface with CD40 on DCs increases DC ability to activate CD8+ T cells and generate memory[14–16]. Several studies, nonetheless, show that CD40–CD40L interaction is not necessary to generate memory CD8+ T cells after bacterial infection[17,18].

Conventional αβ CD4+ T cells can capture bacteria efficiently from DCs through a process termed transinfection[19]. As this activity is driven by T cells and not by bacteria[19], it might be more appropriately termed transphagocytosis. Transphagocytic CD4+ (tpCD4+) T cells kill internalized bacteria in lysosomes, in a manner reminiscent of innate immune cells, thus contributing to the early immune response[19], an effect that raises the question as to whether tpCD4+ T cells are true APCs. Human T cells have been shown to present soluble antigens and activate other T cells in vitro[20,21], although in physiological situations these cells are not considered APCs due to an incapacity to capture antigen. Here we show that conventional αβ tpCD4+ T cells are true APCs, able to prime and activate naive CD8+ T cells efficiently, and to generate central memory CD8+ T cells, a population directly involved in antitumour activity[22]. These findings expand our knowledge of CD4+ T-cell functions and further blur the division between innate and adaptive immunity. To test whether these functions of tpCD4+ T cells, together with their proinflammatory functions[19], are involved in tumour elimination, we use an aggressive melanoma model as a proof-of-concept. We show that tpCD4+ T cells are immunoprotective in mice, highlighting the potential of tpCD4+ T cells for cancer immunotherapy.

## Results

**tpCD4+ T cells cross-prime naive CD8+ T cells**. tpCD4+ T cells were generated as described[19]. Briefly, αβ CD4+ T cells from OT-II transgenic mouse were co-cultured with bone marrow-derived DC (BM-DC) infected with *L. monocytogenes* expressing ovoalbumin (Listeria-OVA)[23] or with its isogenic wild type strain (Listeria-WT) and decorated with OVAp-II antigen (to increase transphagocytosis[19]). Infected DC-T-cell conjugates were allowed to form (24 h), after which CD4+ T cells (tpCD4+ T cells) were repurified by cell sorting (Supplementary Fig. 1A). tpCD4+ T cells that captured *L. monocytogenes* after 2 h conjugate formation are shown in Supplementary Fig. 1B; quantification of bacteria capture/destruction was reported[19]. To test the antigen-presenting capacity of tpCD4+ T cells, we incubated them with naïve CD8+ T cells isolated from OT-I transgenic mice, which recognize an ovoalbumin peptide (OVAp-I 256–264; SIINFEKL) in the context of the H-2K$^b$ MHC-I haplotype. Western blot confirmed that Listeria-OVA, but not Listeria-WT, expressed OVA (Supplementary Fig. 1C). Flow cytometry analysis of CD8+ T cells (Fig 1a; Supplementary Fig. 1D) showed strong CD8+ T-cell proliferation as measured by CellTrace Violet dilution[24] by 2 days after exposure to tpCD4+ T cells, only in cells in contact with tpCD4+ T cells that had captured Listeria-OVA (hereafter Listeria-OVA tpCD4+ T cells). This extremely proliferative population expressed high CD8 levels (Fig. 1a) and showed blast morphology (Supplementary Fig. 1E). CD8+ T cells incubated with Listeria-WT tpCD4+ T cells nonetheless did not proliferate (Fig. 1a). We detected expression of T-cell activation markers CD69 and CD25 on CD8+ T cells incubated with Listeria-OVA tpCD4+ T cells, but not with Listeria-WT tpCD4+ T cells (Fig. 1b). Listeria-OVA tpCD4+ T cells induced levels of CD8+ T-cell proliferation similar to those generated by BM-DC loaded with soluble OVAp-I or by polyclonal T-cell activation by CD3/CD28 antibodies. BM-DC infected with Listeria-OVA induced low levels of CD8+ T-cell proliferation (Fig. 1c; Supplementary Fig. 1F). Activation measured as CD25 expression was similar in Listeria-OVA and Listeria-WT tpCD4+ T cells, which confirmed that CD8+ T-cell activation was due to the antigen presentation ability of tpCD4+ T cells and not to a difference in activation state (Supplementary Fig. 1G). Listeria-OVA tpCD4+ T cells did not promote proliferation of WT mouse CD8+ T cells that do not recognize OVAp-I (Fig. 1d), but potently induced proliferation of CD8+ T cells from OT-I mice (Fig. 1a, c, d), which confirms antigen specificity. The flow cytometry gate strategy is shown in Supplementary Fig. 1D.

**tpCD4+ T cells process bacterial antigens**. Although we did not detect OVA in the extracellular medium (Supplementary Fig. 1C), Listeria-OVA might secrete trace amounts of soluble OVA. To determine whether CD4+ T cells capture OVA from the medium (soluble or exosome-associated) or associated to bacteria by transphagocytosis, we allowed CD4+ T cells to contact infected DC, or blocked DC-T-cell contact with a polycarbonate barrier (transwell) that impedes cell interaction but allows passage of soluble material and exosomes (Supplementary Fig. 2A). Cell-sorted repurified CD4+ T-cell populations were then used as APC to stimulate naïve OT-I CD8+ T cells. Only CD4+ T cells that contacted infected DC directly induced CD8+ T-cell proliferation (Supplementary Fig. 2B). Neither naïve nor activated CD4+ T cells barrier-blocked from contact with infected DC activated CD8+ T cells (Supplementary Fig. 2B), which indicates that tpCD4+ T cells captured OVA as bacteria-associated antigen (Fig. 1). To further confirm that OVA presentation result from bacterial processing inside tpCD4+ T cells, we generated another strain of Listeria-OVA totally impeded in OVA secretion (Listeria-OVA2; Supplementary Fig. 2C). tpCD4+ T cells capturing Listeria-OVA2 potently induced CD8+ T-cell proliferation (Supplementary Fig. 2D).

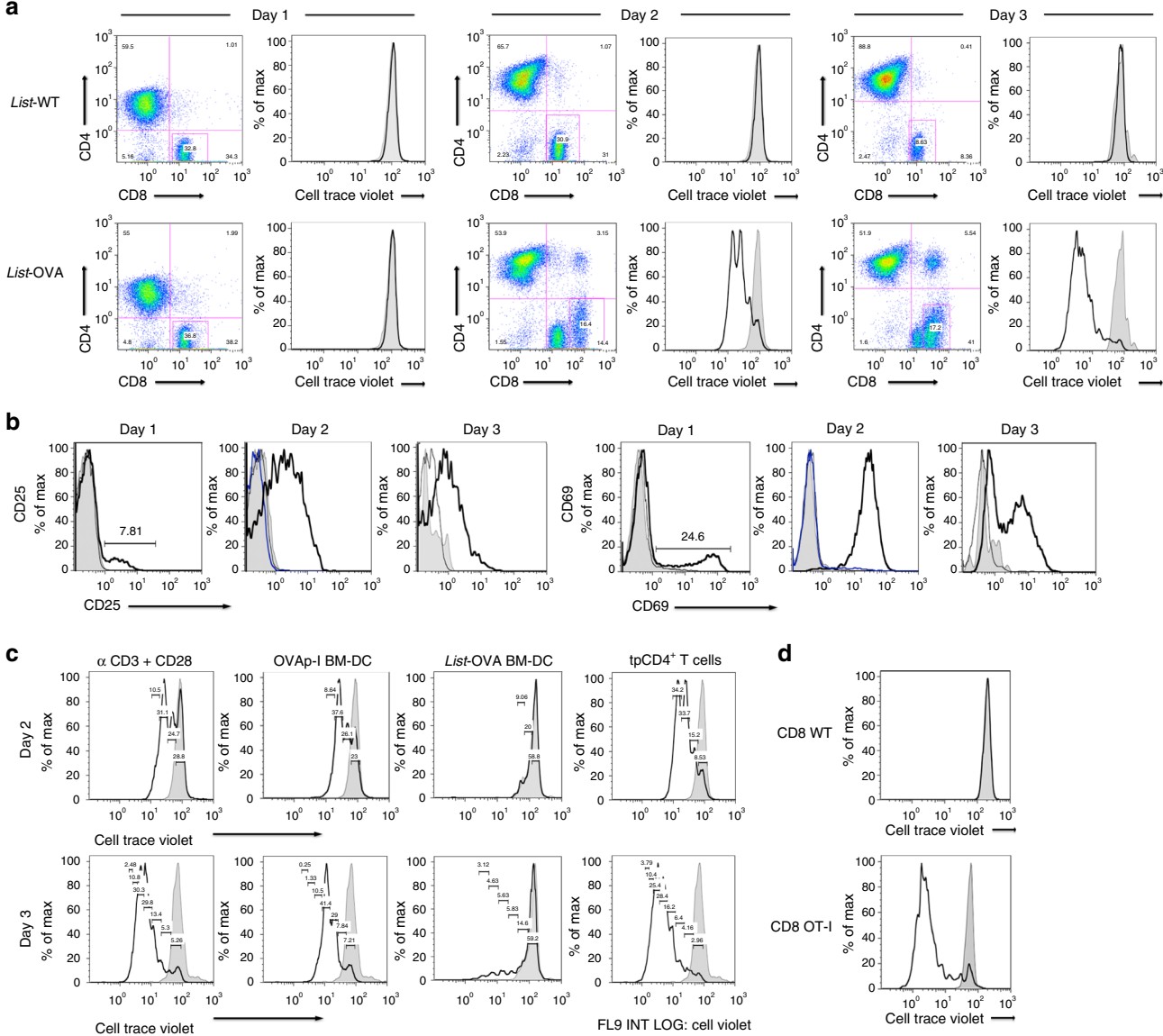

**Fig. 1** tpCD4+ T cells prime naive CD8+ T cells. **a** OT-I naive CD8+ T-cell proliferation was measured by CellTrace Violet staining at 1, 2 or 3 days after contact with Listeria-WT (top) or Listeria-OVA (bottom) tpCD4+ T cells. Non-activated naïve CD8+ T cells are shown in grey. FACS gating is indicating in Supplementary Fig. 1D. **b** CD25 or CD69 expression by CD8+ T cells incubated with Listeria-WT (thin black line) or Listeria-OVA (thick black line) tpCD4+ T cells. Non-proliferative CD8low (blue), non-activated CD8+ T cells (grey). **c** Proliferation of CD8+ T cells activated by anti-CD3/CD28 antibodies, by OVAp-I-loaded BM-DC, by Listeria-OVA BM-DC, or by Listeria-OVA tpCD4+ T cells, on days 2 (top) or 3 (bottom) post-activation. Non-activated CD8+ T cells are shown (grey). **d** Proliferation of CD8+ T cells from C57BL/6-WT or OT-I mice after conjugation with Listeria-OVA tpCD4+ T cells. Non-activated CD8+ T cells are shown (grey)

tpCD4+ T-cell-mediated antigen presentation to CD8+ T cells might involve endogenous processing of bacteria by tpCD4+ T cells, or might be due to capture of MHC/antigen molecules from the DC surface[25]. To discriminate between these possibilities, we incubated CD4+ T cells with Listeria-WT- or Listeria-OVA-infected BM-DC loaded with soluble OVAp-I. We used the purified tpCD4+ T cells to stimulate naïve CD8+ T cells. If the mechanism involves MHC/antigen transfer or exogenous antigen peptide transfer from DC to CD4+ T cells, antigen capture from DC would activate CD8+ T cells similarly in both conditions (Supplementary Fig. 2E). We found that Listeria-OVA but not Listeria-WT tpCD4+ T cells strongly activated CD8+ T cells (Fig. 2a), which indicates that CD8+ T-cell activation was due principally to antigen processing by tpCD4+ T cells. To confirm this finding, we used tpCD4+ T cells from AND transgenic mice

(moth cytochrome c (MCC) peptide-specific TCR) that express H-2Kk alone, or on a mixed background that co-expresses H-2Kk/ H-2Kb. H-2Kk/b-expressing cells induced strong CD8+ OT-I T-cell proliferation compared to those that expressed H-2Kk/k; this indicated that antigen presented by MHC-I in tpCD4+ T cells derived mainly from intracellular processing (Fig. 2b), and only a minor part by capturing MHC/antigen complexes from DC.

To exclude a contribution to antigen presentation by post-purification DC contamination (transphagocytosis occurs by close contact between CD4+ T cells and DC), we incubated OT-I CD8+ T cells with H-2Kk/k or H-2Kk/b tpCD4+ T cells transphagocytosing Listeria-OVA from infected H-2Kk/k BM-DC (unable to activate CD8+ T cells from OT-I mice). Only -H-2Kk/b tpCD4+ T cells induced CD8+ T-cell proliferation (Fig. 2c) and were stained with 25D1.16 antibody specific for the

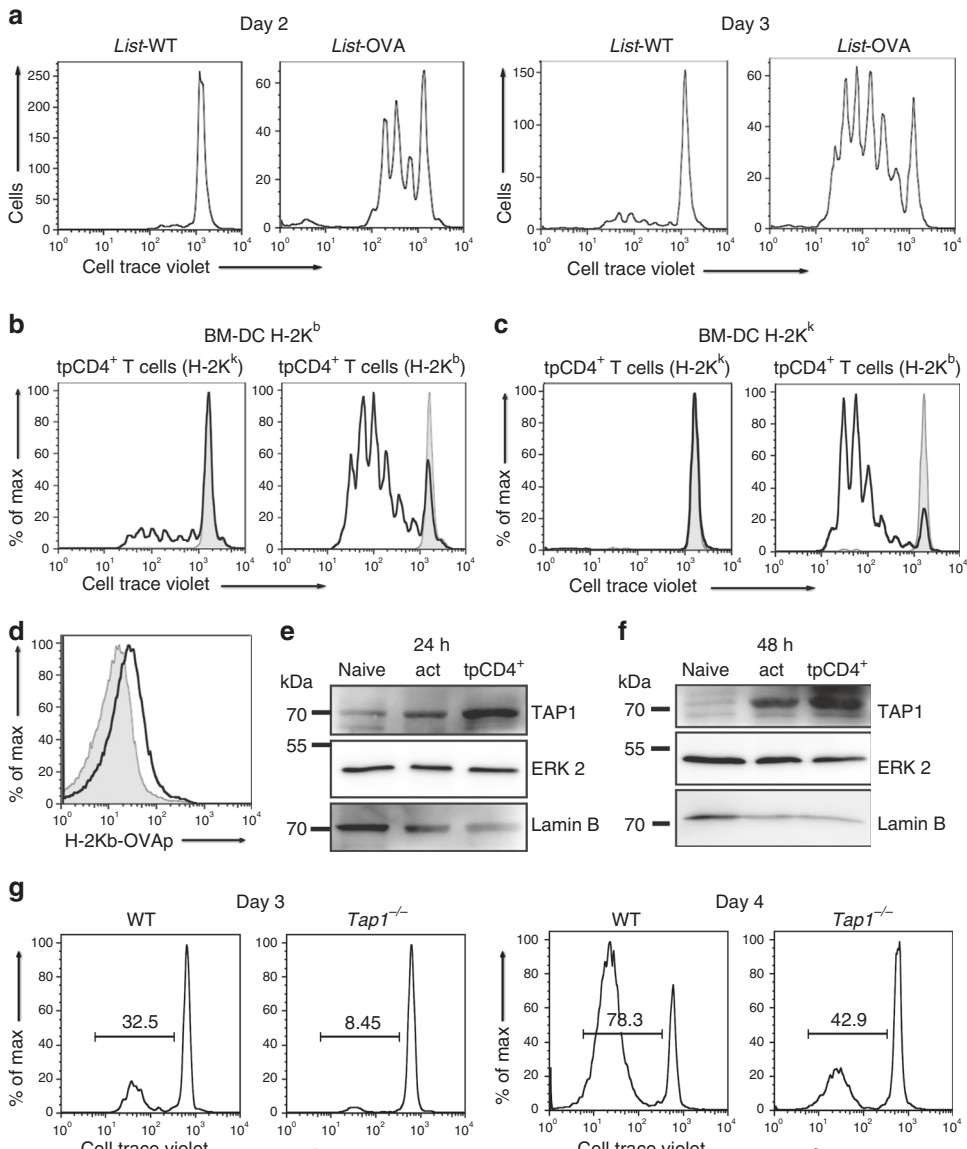

**Fig. 2** tpCD4$^+$ T cells process bacterial antigens. **a** Proliferation of OT-I CD8$^+$ T cells, 2 or 3 days after incubation with Listeria-WT or Listeria-OVA tpCD4$^+$ T cells that captured bacteria from OVAp-I-loaded BM-DC. **b** Proliferation of OT-I CD8$^+$ T cells incubated with Listeria-OVA tpCD4$^+$ T cells (H-2K$^{k/k}$ or H-2K$^{k/b}$), which captured bacteria from infected H-2K$^b$ BM-DC, or with Listeria-WT tpCD4$^+$ T cells (grey). **c** as for **b**, except transphagocytosis was carried out using infected H-2K$^{k/k}$ BM-DC (48 h). **d** H-2K$^b$/OVA expression (detected with anti-OVAp-I/H-2K$^b$ antibody) by Listeria-OVA tpCD4$^+$ T cells (H-2K$^{k/b}$, *black* line; H-2K$^{k/k}$, *grey*) that captured bacteria from H-2K$^{k/k}$ BM-DC. **e, f** Western blot showing TAP1 expression in naive, activated (*act*, by OVAp-I-loaded BM-DC) CD4$^+$ T cells, or Listeria-OVA tpCD4$^+$ T cells at 24 **e** or 48 h **f** after activation. ERK2 and laminB were used as protein loading controls. Note that all original western blots together with the membranes stained for molecular size markers are provided at the end of the manuscript in Supplementary Fig. 5. **g** Proliferation of CD8$^+$ T cells from OT-I mice, 3 or 4 days after conjugation with Listeria-OVA tpCD4$^+$ T cells (from *Tap1$^{-/-}$* or isogenic WT mice)

H-2K$^b$/OVAp-I complex (Fig. 2d). These experiments using DC unable to activate OT-I CD8$^+$ T cells thus excluded DC contamination as a cause of CD8$^+$ T-cell activation, and confirmed that tpCD4$^+$ T-cell-mediated antigen presentation was derived from intracellular processing.

To further test the antigen presentation capacity of CD4$^+$ T cells after bacterial capture, we analyzed the antigen presentation machinery in tpCD4$^+$ T cells. In Western blot analysis, tpCD4$^+$ T cells showed increased expression of TAP1 (Fig. 2e, f), a key protein in antigen presentation on MHC-I[26,27]. Antigen presentation after bacterial capture by tpCD4$^+$ T cells from *Tap1$^{-/-}$* mice was greatly limited (Fig. 2g), as tpCD4$^+$-induced CD8$^+$ T-cell proliferation was reduced when

*Tap1$^{-/-}$* mouse CD4$^+$ T cells were used as APC. The data suggest that tpCD4$^+$ T cells might use, at least in part, the canonical crosspresentation pathway described for DC.

**tpCD4$^+$ T cells form immune synapses with naive CD8$^+$ T cells**. Listeria-OVA transphagocytosis by CD4$^+$ T cells led to increased H-2K$^b$ expression, accumulation of H-2K$^b$ coupled to OVAp-I antigen (of bacterial origin), and expression of costimulatory receptor ligands such as CD86[28] (Fig. 3a–c); these results are compatible with antigen presentation through the MHC-I pathway. OVAp-I expression on H-2K$^b$ tpCD4$^+$ T cells was detected 48 h post-transphagocytosis (Fig. 3c), which corroborates intracellular processing of internalized bacteria. The ensuing activation

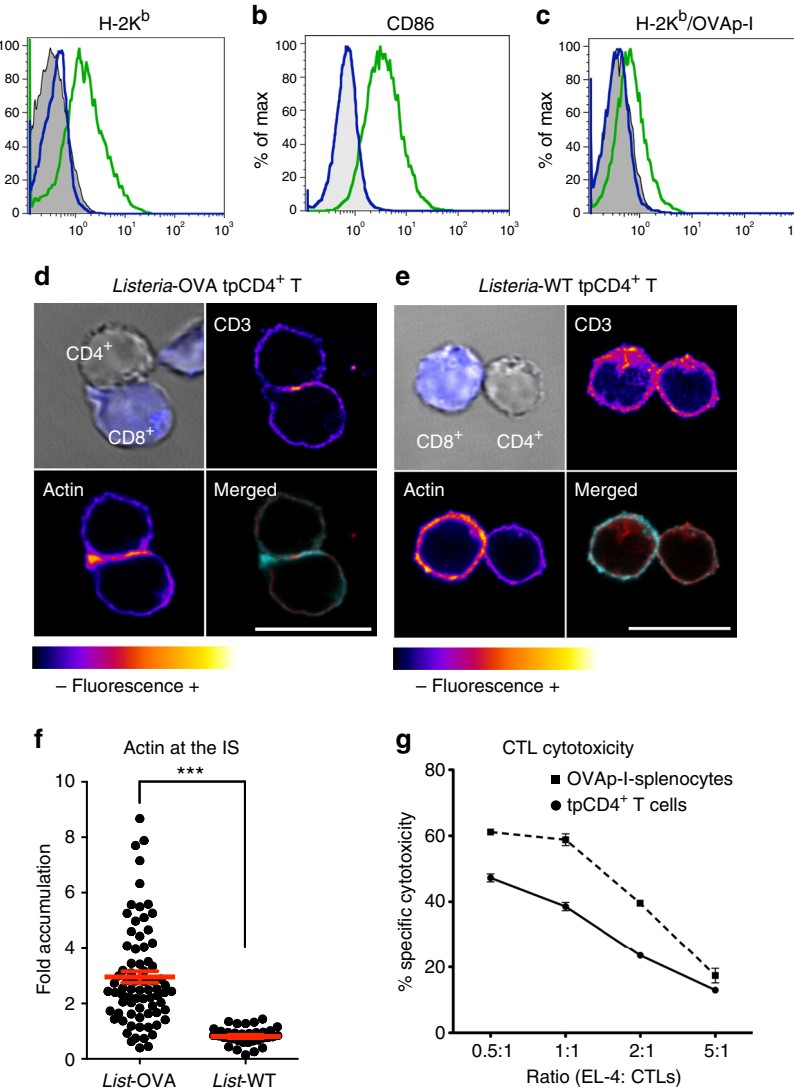

**Fig. 3** tpCD4+ T cells establish immune synapses with naive CD8+ T cells. H-2K^b (**a**) or CD86 (**b**) expression on CD4+ T cells before (blue line) and after Listeria-OVA transphagocytosis (green line). Antibody negative controls are shown (grey). **c** H-2K^b/OVA expression on Listeria-OVA tpCD4+ T cells, 24 (blue) or 48 h (green) post-transphagocytosis and on non-transphagocytic T cells (grey). Confocal images of Listeria-OVA (**d**) or Listeria-WT tpCD4+ T cells (**e**) incubated 1 h with OT-I naive CD8+ T cells (blue). CD3 and actin fluorescence is shown on a "fire" scale. Merged images show actin (cyan) and CD3 (red). Bars = 10 μm. **f** Quantification of actin accumulation at the IS analyzed using *Synapse measures* software[31]. Naïve OT-I CD8+ T cells were conjugated with Listeria-OVA or Listeria WT tpCD4+ T cells and fluorescence microscopy was performed as in **d** and **e**. Each dot represents the fold increase of actin accumulation in the IS in an individual CD8+/tpCD4+ T cell contact from three independent experiments, the mean and the s.e.m. are shown in red. ***P < 0.001 (analyzed by Mann–Whitney U-test). **g** Relative specific cytotoxicity of effector CD8+ T cells (CTL) activated by Listeria-OVA tpCD4+ T cells (solid line) or by OVAp-I-loaded splenocytes (dashed line). Various EL-4 (target cell):CTL ratios were measured

of CD8+ T cells appeared to involve formation of CD4+/CD8+ T-cell conjugates (Supplementary Fig. 1e), indicative of IS generation, a hallmark of T-cell activation by APC[6,29]. The mature IS central zone (cSMAC) bears MHC-TCR complexes, and the peripheral zone (pSMAC) is a ring-like structure of adhesion molecules and F-actin[6,30,31]. Immunofluorescence analysis confirmed that naïve CD8+ T cells formed mature IS after exposure to Listeria-OVA tpCD4+ T cells (Fig. 3d, Supplementary Movie 1). These tpCD4+/CD8+ T-cell conjugates showed a TCR complex in the cSMAC and accumulation of polymerized actin in the pSMAC. In the infrequent cases that Listeria-WT tpCD4+ T cells formed conjugates with CD8+ T cells, we found no evidence of such SMAC structures (Fig. 3e), nor actin accumulation at the IS (Fig. 3f). Quantification of actin accumulation at the IS showed a threefold increase in the case of Listeria-OVA tpCD4+ T cells as APC (Fig. 3f).

We also tested the cytotoxic capacity of CD8+ T cells and found that their activation by antigen-presenting Listeria-OVA tpCD4+ T cells enabled them to eradicate OVAp-I-expressing EL-4 lymphoma target cells[32] (Fig. 3g). The in vitro data thus show that tpCD4+ T cells are professional APC that capture bacteria, degrade them and activate naïve CD8+ T cells to induce a cytotoxic response.

**tpCD4+ T cells prime naive CD8+ T cells in vivo**. To test whether tpCD4+ T cells that had transphagocytosed bacteria in vitro also activate naïve CD8+ T cells in vivo, we isolated naïve CD8+ T cells from OT-I mice expressing the CD45.1 allele marker, stained them ex vivo with a cell proliferation marker, and transferred them with tpCD4+ T cells into C57BL/6/CD45.2+ recipient mice. Spleen CD8+CD45.1+ T cells proliferated in

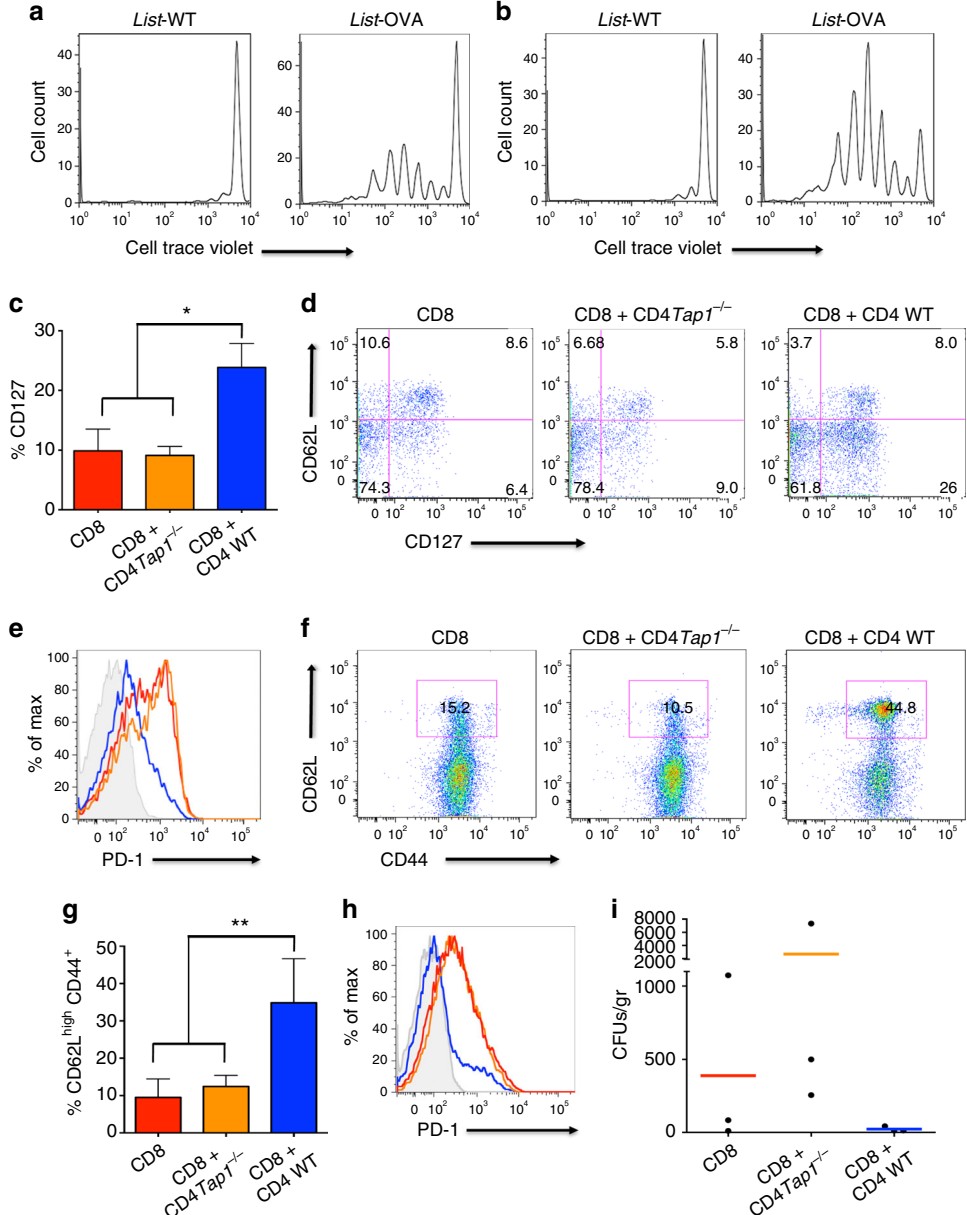

**Fig. 4** tpCD4[+] T cells-mediated antigen presentation generates CD8[+] T central memory. **a, b** In vivo proliferation of CellTrace Violet-stained OT-I CD8[+]CD45.1[+] T cells, injected into recipient C57BL/6 CD45.2[+] mice. Listeria-WT or Listeria-OVA tpCD4[+] T cells ($2 \times 10^6$, **a**; $5 \times 10^6$, **b**) were adoptively transferred. Proliferation was analyzed 3 days post-transfer. **c–h**, Analysis of memory CD8[+] T cell generation in vivo. $Rag1^{-/-}$ mice were adoptively transferred i.v. with WT CD8[+] T cells (group 1), $Tap1^{-/-}$ CD4[+] plus WT CD8[+] T cells (group 2), and WT CD4[+] plus WT CD8[+] T cells (group 3). The following day, all mice were challenged with *L. monocytogenes* ($10^3$ bacteria/ per mice; i.v.). At 12 (**c–e**) or 30 days (**f–h**) after challenge, the CD8[+] T-cell population from spleen was analyzed. FACS gating is indicating in Supplementary Fig. 4B. **c** Percentage of CD127[+], gated on CD3[+]CD8[+]CD44[+]CD62L[−], pre-memory T cells. Data shown as mean ± s.d. for 3 mice per group (group 1, red; group 2, *orange*; group 3, blue). *$P < 0.05$ analyzed by ANOVA and Bonferroni post-test. **d** Representative dot plots showing CD62L and CD127 staining from the CD3[+]CD8[+]CD44[+] T-cell population. **e** Representative histogram showing PD-1 levels on the CD3[+]CD8[+]CD44[+] T-cell population (group 1, red; group 2, orange; group 3, blue; Fluorescence Minus One (FMO) control, grey). **f** Representative dot plots showing CD62L and CD44 staining gated on the CD3[+]CD8[+] T-cell population. The CD62L[high] population was also positive for CD127 (not shown). **g** Percentage of CD62L[high]CD44[+] gated on cells indicated in **f** (red outline). Data shown as mean ± s.d. for 4 mice per group (group 1, red; group 2, orange; group 3, blue). *$P < 0.05$ analyzed by ANOVA and Bonferroni post-test. **h** Representative histogram showing the amount of PD-1 on the CD3[+]CD8[+]CD44[+] T-cell population (group 1, red; group 2, orange; group 3, blue; FMO control, grey). **i** *L. monocytogenes* load in spleen 2 days after a second bacterial challenge (50 days after the first). Each dot represents one mouse (3 mice per group), the mean is shown in red (group 1), orange (group 2) and blue (group 3)

response to Listeria-OVA but not to Listeria-WT tpCD4[+] T cells (Fig. 4a, b). FACS gate strategy followed in this figure was the same than this indicated in Supplementary Fig. 1D; once excluded the debris, we gated in live cells. Lymphocytes were analyzed by antibody staining for CD45.1[+]CD8[+].

To confirm that CD4[+] T cells capture bacteria and present antigens in vivo in the course of a bacterial infection, irradiated recipient C57BL/6 mice (H-2K[b]) received a H-2K[k] bone marrow stem cell progenitor transplant. The APC of the reconstituted mice cannot prime OT-I CD8[+] T cells. After 1 month, AND

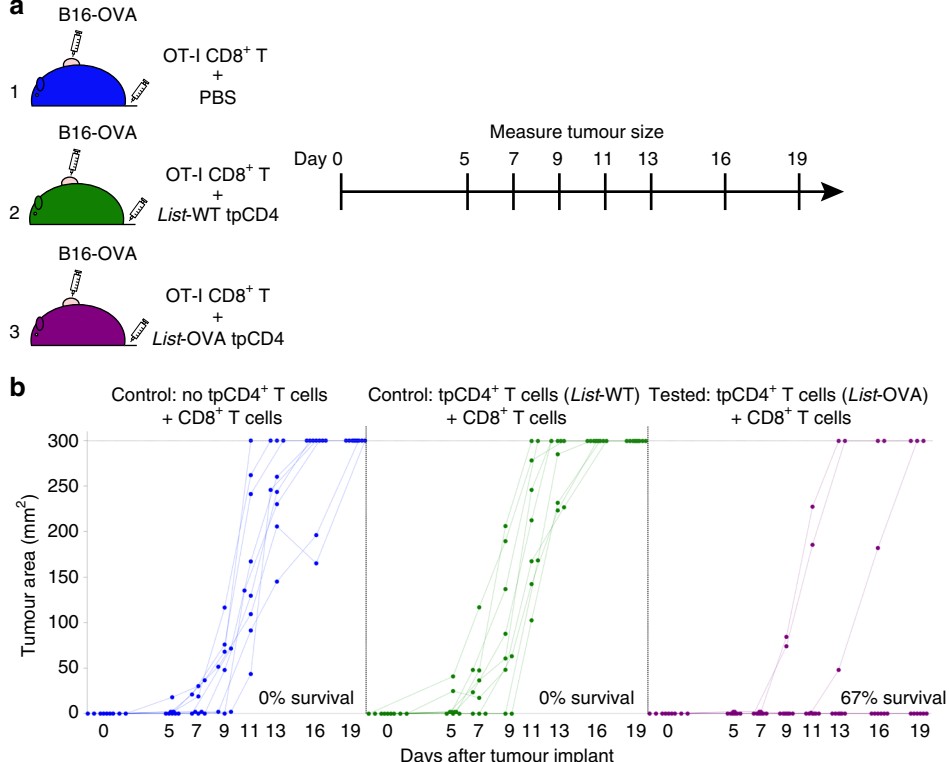

**Fig. 5** tpCD4[+] T cells protect from melanoma. **a** Scheme, and (**b**) results of the proof-of-concept experiment to test tpCD4[+] T-cell ability to activate CD8[+] T cells that recognize and eliminate the tumour. OVA-expressing B16 melanoma cells ($5 \times 10^5$) were injected s.c. in the mid-right flank of recipient mice. All groups (9 mice per group) were adoptively transferred i.v. with naive CD8[+] T cells from OT-I mice, together with PBS (group 1, blue), Listeria-WT tpCD4[+] T cells (group 2, green) or Listeria-OVA tpCD4[+] T cells (group 3, magenta). Tumours were monitored every 2–3 days. In **b**, each dot represents tumour size in one mouse measured at the indicated time. Mice with tumours ≥ 300 mm[2] were killed to avoid unnecessary suffering. The percentage of survival is indicated

CD4[+] T cells (H-2K[k/b] or H-2K[k/k]) were adoptively transferred into the recipient mice, together with OT-I CD8[+]CD45.1[+] T cells and MCC peptide (to promote bacteria capture by CD4[+] T cells[19]; Supplementary Fig 3a). Lack of H-2K[b] haplotype in the myeloid lineage was confirmed in recipient mice (Supplementary. Fig. 3b), which were challenged with Listeria-OVA. Only mice that received H-2K[b] CD4[+] T cells activated OT-I CD8[+]CD45.1[+] T cells (Supplementary. Fig. 3c–e), which confirmed tpCD4[+] T-cell-mediated transphagocytosis and antigen presentation in the course of an in vivo bacterial infection.

**tpCD4[+] T cells induce CD8[+] T central memory during infection.** To analyze the role of antigen presentation by tpCD4[+] T cells in physiological conditions in the context of endogenous WT CD8[+] and CD4[+] T-cell repertoire (not OVA antigen-specific), we performed an adoptive transfer experiment (summarized in Supplementary Fig. 4a). Rag1[−/−] recipient mice (lymphocyte-deficient) were divided into three groups. One group received CD4[+] T cells from WT C57BL/6 mice, one received CD4[+] T cells from Tap1[−/−] mice (antigen presentation-defective; Fig. 2g), and one received no CD4[+] T cells. The same day, all mice also received naive CD8[+] T cells from WT C57BL/6 mice, and the following day all mice were infected with a low dose of L. monocytogenes (10[3] bacteria per mouse). Mice from each group were killed at day 5, 12 and 30 post-challenge and the spleen CD8[+] T-cell phenotype was analyzed by flow cytometry (Supplementary Fig. 4b). At day 5 post-challenge, we detected no differences in the CD8[+] T-cell populations from all groups (not shown). At 12 days after challenge, the group that received

WT CD4[+] T cells showed three times more CD3[+]CD8[+] CD44[+]CD62L[−]CD127[+] T cells than the others (Fig. 4c,d). CD127, the IL-7 receptor, is described as a CD8[+] memory T-cell marker[33,34]. These data indicate that tpCD4[+] T-cell-mediated antigen presentation has a major role in the generation of pre-memory CD8[+] T cells. PD-1 expression in pre-memory CD8[+] T cells was greatly reduced in the WT CD4[+] T-cell recipient group (Fig. 4e). CD40L expression was similar in polyclonal activated CD4[+] T cells from WT and Tap1[−/−] mice (Supplementary Fig. 4c). Tap1[−/−] and WT CD4[+] T cells secrete similar IFN-γ levels in the presence of IFN-γ[+] natural killer (NK) cells[35], which is the case of Rag1[−/−] mice[36,37]; this indicated that the differences between mouse groups that received WT or Tap1[−/−] CD4[+] T cells are due to a difference in antigen presentation ability. Results at 30 days after bacterial challenge were similar; the group transferred with WT CD4[+] T cells showed larger numbers (3–4 times) of central memory CD8[+] T cells (CD3[+] CD8[+]CD44[+]CD62L[high]CCR7[+]CD127[+]) than the other groups (Fig. 4f, g). Levels of CCR-7, expressed by central memory CD8[+] T cells[38], was low in CD8[+]CD44[+]CD62L[high] cells in mice that received Tap1[−/−] CD4[+] T cells vs. mice that received WT CD4[+] T cells (Supplementary Fig. 4D). PD-1 expression on CD3[+] CD8[+]CD44[+] T cells in the group that received WT CD4[+] T cells was also lower than the other groups (Fig. 4h), which indicated that memory CD8[+] T cells generated by CD4[+] T-cell antigen presentation might be resistant to exhaustion. Like the group that received CD8[+] but not CD4[+] T cells, the group that received Tap1[−/−] CD4[+] T cells had fewer central memory T cells and expressed high PD-1 levels, which again indicated that tpCD4[+] T cells have a direct role in CD8[+] T-cell memory generation due

to their antigen presentation ability and not to indirect DC stimulation.

To analyze the cytotoxic response of these in vivo-generated memory CD8$^+$ T cells, we rechallenged mice from all three groups (50 days after first infection) with LD50 *L. monocytogenes* ($5 \times 10^4$ bacteria per mouse); 2 days later, spleen bacterial load was quantified by colony counting (Supplementary Fig. 4). The group transferred with WT CD4$^+$ T cells eliminated more bacteria (Fig. 4i), which indicated that memory CD8$^+$ T cells generated by CD4$^+$ T-cell antigen presentation also cleared bacteria efficiently during secondary infection.

To further confirm in vivo the role of tpCD4$^+$ T-cell-dependent antigen presentation in the context of a robust innate immune response, we used *Rag1$^{-/-}$* receptor mice transferred with WT CD8$^+$ T cells together with WT or *B2m$^{-/-}$* (beta-2 microglobulin)-deficient CD4$^+$ T cells. *B2m$^{-/-}$* cells expressed low MHC-I levels[39]. Receptor mice were challenged with Listeria-OVA2 after cellular transfer and 30 days later, the CD8$^+$ T-cell population was analyzed. Central memory (CD44$^+$CD62L$^{high}$CD8$^+$) generation was reduced in mice transferred with *B2m$^{-/-}$* CD4$^+$ T cells (Supplementary Fig. 4E, F), confirming that antigen presentation via tpCD4$^+$ T cell plays a major role in the generation of central memory. Moreover, the amount of CD8$^+$ T cells recognizing OVAp-I was greatly increased in the presence of WT tpCD4$^+$ T-cell compared to *B2m$^{-/-}$* CD4$^+$ T cells as revealed by using a dextramer that binds to H-2K$^b$/SIINFEKL-specific CD8$^+$ T cells (Supplementary Fig. 4G).

**Antitumour immunotherapy potential of tpCD4$^+$ T cells**. We observed that transphagocytosis converts conventional αβ CD4$^+$ T cells into potent APC, that generate cytotoxic and memory CD8$^+$ T cells with low PD-1 levels. Together, with the highly pro-inflammatory profile of tpCD4$^+$ T cells[19], this provided a base on which to develop antitumour immunotherapeutic strategies[9,40]. To explore the therapeutic potential of tpCD4$^+$ T cells as APC, we tested whether tpCD4$^+$ T-cell-mediated activation of naive CD8$^+$ T cells confers protection from tumour formation in a murine model of malignant melanoma. We inoculated the murine melanoma cell line B16-OVA[41–43] subcutaneously into C57BL/6 mice. The same day, naive CD8$^+$ T cells from OT-I mice were adoptively transferred by a single intravenous injection with PBS (group 1), Listeria-WT tpCD4$^+$ T cells (group 2), or Listeria-OVA tpCD4$^+$ T cells (group 3; Fig 5a). All control mice treated with vehicle (PBS) or Listeria-WT tpCD4$^+$ T cells developed tumours within 11 days of B16-OVA injection (Fig. 5b). Administration of Listeria-OVA tpCD4$^+$ T cells prevented tumour formation in 6 of 9 mice and delayed tumour formation in one mouse; vaccination with antigen-presenting tpCD4$^+$ T cells thus conferred protection against melanoma in 7 of 9 cases (Fig. 5b). These results of this proof-of-concept experiment show that cross-presentation by tpCD4$^+$ T cells can be used in vivo to prime CD8$^+$ effector T cells against tumours.

**Discussion**

Here we show that presentation of bacterial antigens activating naïve CD8$^+$ T cells, thought to be limited to professional APC of the innate immune system[44], are recapitulated by conventional CD4$^+$ T cells, considered the paradigm of adaptive immune response cells. This finding diffuse the traditional boundary between innate and adaptive immunity, as do other specialized, non-conventional lymphocyte populations[45]. Our data suggest that tpCD4$^+$ T-cell-mediated activation of CD8$^+$ T cells could take place though cross-presentation; OVA antigen was produced by bacteria (exogenous generation of antigen) and it was not

secreted (Supplementary Fig. 2A–D), Listeria degradation by tpCD4$^+$ T cells occurs in a lysosomal compartment[19], and antigen presentation occurs (at least partially) through the TAP1 pathway, the canonical cross presentation pathway in DC (Fig. 2g). On the other hand, bacterial escape from the endosome into the cytosol is a normal phase of the Listeria life cycle and we cannot exclude bacterial degradation in the cytosol. Therefore, more experiments were needed to precisely determine whether tpCD4$^+$ T-cell-mediated antigen presentation occurs through cross-presentation.

These antigen-presenting abilities of conventional CD4$^+$ T cells could have a role in secondary immune responses, in which large numbers of CD4$^+$ and CD8$^+$ T cells respond to a given pathogen. The ability of T lymphocytes to present soluble antigens and activate other T cells has been described[20,21], but these studies did not show a role for T-cell-dependent antigen presentation in physiological situations. T-cell incapacity to capture antigens was thought to render them unsuitable as APC in vivo[20]. We showed that CD4$^+$ T cells can capture different bacteria by transphagocytosis and destroy captured pathogens[19]. Here we show that tpCD4$^+$ T cells process and present bacterial peptides in the context of their own MHC to induce potent activation of CD8$^+$ T cells. We found that the vast majority of this CD8$^+$ T-cell activation was due to antigen processing within tpCD4$^+$ T cells, and only a small part to MHC/antigen complexes acquired from DC during DC/T-cell contact. Coinciding with these findings, a microscopy-based study showed that in the course of viral infection, CD8$^+$ T cells are activated after CD4$^+$ T-cell contact with infected DC. Early post-infection events included DC/CD4$^+$ T cell contacts and rarely, DC/CD8$^+$ T-cell contacts; CD4$^+$/CD8$^+$ T-cell contacts were the predominant later events[46].

We show that CD4$^+$ T-cell-dependent antigen presentation generates central memory CD8$^+$ T cells (CD44$^+$CD62L$^{high}$ CCR7$^+$) in vivo in a natural repertoire environment, in the course of a bacterial infection. tpCD4$^+$ T-cell-induced memory CD8$^+$ T cells expressed lower PD-1 levels than those activated by professional phagocytes, mainly DC. These observations expand our knowledge of the roles of CD4$^+$ T cells during CD8$^+$ T-cell memory generation. CD4$^+$ T cells are necessary for CD8$^+$ T-cell memory generation following acute pathogen infection[14], but the mechanisms involved remain unclear. Interaction of CD40L on the CD4$^+$ T-cell surface with CD40 on DC is thought to increase DC ability to activate CD8$^+$ T cells and generate memory[14–16]. Other studies nonetheless show that CD40-CD40L interaction is not necessary to generate memory CD8$^+$ T cells after bacterial infection[17,18]. In all cases, it was thought that antigen is presented to CD8$^+$ T cells exclusively by DC; here we show that CD4$^+$ T cells can present bacterial antigens directly to CD8$^+$ T cells and promote a memory response. It remains to be determined whether tpCD4$^+$ T cells that capture other bacteria (different to Listeria) would be more effective in antigen presentation and the role of bacterial PAMPs in this process.

The ability of tpCD4$^+$ T cells to strongly activate cytotoxic CD8$^+$ T cells with low PD-1 levels, together with their vigorous pro-inflammatory nature[19], could bypass the immunosuppressive envinronment of solid tumours[47,48], which might be exploited in vivo to activate naive CD8$^+$ T cells against tumours. Despite the success of antibody-based therapy against tumour checkpoint blockade, > 50% of cancer patients fail to respond[49]. The advent of new technologies that improve the ability to detect tumour antigens enhance possibilities to attack tumours using novel immunotherapies[50,51], for example by combining these methods with effective antigen presentation systems[8]. We used a proof-of-concept experiment to show that tpCD4$^+$ T cells are effective APC and impede implantation of the aggressive B16-OVA melanoma. A single tpCD4$^+$ T-cell injection protected against the

B16 tumour (Fig. 5b), whereas DC-based vaccination requires several injections for similar protection[52]; this highlights the potent priming activity of tpCD4[+] T cells. These data pave the way for further research on the role of conventional CD4[+] T cells in innate immunity and their interactions with other immune cells, as well as their potential use for tumour immunotherapy.

## Methods

**Mice.** Wild-type C57BL/6 mice, as well as C57BL/6-Tg (TcraTcrb)425Cbn/J OT-II mice expressing a T-cell receptor (TCR) specific for OVA peptide 323–339 in the context of MHC class II (I-A[b])[53], and C57BL/6-Tg(TcraTcrb)1100Mjb/J OT-I mice expressing TCR specific for OVA peptide 257–264 in the context of H-2K[b] [54,55] were purchased from Jackson Laboratory (004194 and 003831, respectively). AND-TCR transgenic mice that recognize moth cytochrome *c* 88–103 (ANER-ADLIAYLKQATK) (MCCp) on I-E[k], and express H-2K[b], H-2K[k], or both haplo-types have been described[56,57], deficient in MCH-I expression were from Jackson Laboratory. Male or female mice aged 8 to 12 weeks were used for experiments. Mice were maintained in the specific-pathogen-free (SPF) unit at the Universidad Autónoma de Madrid School of Medicine and the Centro Nacional de Biotecnología (CNB, Madrid) animal facilities and for some experiments in the SPF unit at the Centro Nacional de Investigaciones Cardiovasculares (CNIC, Madrid). Sample size chosen was calculated by using "*Gpower3.1.2*", but this was limited by the available mice in some experiments. Experimental groups were assigned randomly and measurements made in a double-blind manner. Experimental procedures were approved by the Committee for Research Ethics of the Universidad Autónoma de Madrid, CNIC and CNB-CSIC, and experiments were conducted in accordance with Spanish and EU guidelines. All procedures were approved by the Madrid local authority (project no PROEX 431/15).

**Bacterial strains.** Bacterial strains used were *Listeria monocytogenes*-OVA (pPL2-LLO-OVA), which expresses OVA protein[23], *Listeria monocytogenes*-OVA2 generated in this work also expressing OVA and their WT isogenic strain *L. monocytogenes* 10403S. To generate Listeria-OVA2 the genes encoding for OVA (with the codon usage optimized for Gram+) and GFP were cloned in the pPL2 insertion plasmid under the pHELP promoter[60] (with no signal peptide to avoid protein secretion) generating pPL2OVA2GFP that was electroporated in *L. monocytogenes* 10403S. The plasmid design allows to easily interchange OVA and GFP by other desired genes. Bacteria were grown in brain-heart infusion (BHI) medium (overnight, 37 °C), diluted, recovered at mid-logarithmic growth phase (OD$_{600nm}$ = 0.5), and washed in PBS before intravenous (i.v.) inoculation.

**Cells.** Bone marrow dendritic cells (DC) were generated as described[19]. Briefly, cells from mouse bone marrow were incubated with recombinant murine granulocyte–macrophage colony-stimulating factor (rm-GM-CSF, 20 ng/ml) for 9 days, changing the medium every 3 days. Phenotypic characteristics were assessed by flow cytometry on day 10 (CD11c[+], IA/IE[+], Gr1[-] to ensure correct differentiation. Maturation was induced with 20 ng/ml lipopolysaccharide (LPS; 24 h). Primary mouse CD4[+] T cells were obtained from single-cell suspensions of lymph nodes (LN) and spleens. Cell suspensions were incubated with biotinylated antibodies to CD8, IgM, B220, CD19, MHC class II (I-A[b]), CD11b, CD11c and DX5, and with streptavidin microbeads. CD4[+] T cells were negatively selected in an auto-MACS Pro Separator (Miltenyi Biotec). To isolate naïve CD8[+] T cells, spleen cell suspensions were incubated with the same biotinylated antibodies, and anti-CD25 and -CD4 antibodies rather than CD8. The EL-4 lymphoma cell line was maintained in RPMI 1640 medium (Fisher Scientific) supplemented with 10% heat-inactivated fetal calf serum (FCS), 0.1 U/ml penicillin, 0.1 mg/ml streptomycin (Lonza) and 0.05 mM 2-mercaptoethanol. The OVA-expressing B16 melanoma cell line was maintained in RPMI 1640 with 0.4 mg/ml geneticin. Antibiotic was removed by washing 48 h before injection into mice.

**Antibodies.** Antibodies to mouse proteins conjugated with different fluorochromes were anti-CD69 (H1.2F3, 1:100 dilution, BD Biosciences), anti-CD25 (PC61.5, 1 : 200 dilution, Tonbo Biosciences), anti-CD4 (GK1.5, 1:100 dilution, BD Biosciences), anti-CD8 (53-6.7, 1:100 dilution BD Biosciences), anti-CD11c (N418, 1:100 dilution, Tonbo Bioscience), anti-MHC_II (IA/IE) (2G9, 1:100 dilution, BD Biosciences), anti-Gr1 (RB6-8C5, 1:200 dilution, Tonbo Biosciences), anti-PD-1 (29 F.1A12, 1:200 dilution, Biolegend), anti-CD127 (A7R34, 1:200 dilution, Tonbo Biosciences), anti-CD44 (IM7, 1:100 dilution, Tonbo Biosciences), anti-CD62L (MEL-14, 1:100 dilution, Tonbo Biosciences), anti-CD45.1 (A20, 1:100 dilution, BD Biosciences), anti-H2Kb (AF6-88.5, 1:100 dilution, BD Biosciences). Biotin-conjugated antibodies for cell isolation were anti-CD4 (GK1.5, 1:250 dilution, ImmunoStep or BD Biosciences), anti-CD8 (53-6.7, 1:250 dilution, BD Biosciences), anti-IgM (B11/7, 1:250 dilution, Immunostep), anti-B220 (RA3-6B2, 1:250 dilution, BD Biosciences), anti-CD19 (ID3, 1:250 dilution, BD Biosciences), anti-MHC-II (IA/IE) (2G9, 1:250 dilution, BD Biosciences), anti-CD11b (M1/70,

1:250 dilution, BD Biosciences), anti-CD11c (HL3, 1:250 dilution, BD Biosciences), anti-CD49b (DX5, 1:250 dilution, BD Biosciences), anti-CD25 (PC61, 1:250 dilution, BD Biosciences). Biotin-conjugated antibodies to label cells for flow cytometry analysis were anti-CCR-7 (4B12, 1:100 dilution, eBioscience), anti-CD45.1 (A20, 1:100 dilution, BD Biosciences). Antibodies against mouse proteins anti-CD16/CD32 (2.4G2, 1:200 dilution, eBioscience), anti-CD45.2 (17A2, 5 ug/ml, eBioscience), anti-CD28 (37.51, 2 ug/ml, BD Pharmingen), anti-TAP-1 (M-18, polyclonal antibody, 1:200 dilution, Santa Cruz), anti-ERK-2 (C-14, polyclonal, 1:200, Santa Cruz), -lamin B (M-20, 1:200 dilution, Santa Cruz). Mouse allophycocyanin-25-D1.16 monoclonal antibody (specific for SIINFEKL/H-2K[b]) was purchased from eBioscience (1/100 dilution). Anti-OVA antibody was provided by Dr. David Sancho (CNIC, Madrid). Anti-Listeria (polyclonal, 1:200 dilution, AbD Serotec). Secondary goat anti-hamster and anti-rabbit antibodies conjugated to AlexaFluor488, 647, or 568 were purchased from Life Technologies (1:200 dilution); horseradish peroxidase (HRP)-anti-goat IgG and anti-rabbit IgG were from Thermo Scientific (1:10 000 dilution).

**Reagents.** OVAp-II (OVA 323–339; ISQAVHAAHAEINEAGR) and OVAp-I (OVA 257–264; SIINFEKL) were generated at the Centro de Biología Molecular Severo Ochoa (CBM, Madrid) and Centro Nacional de Biotecnología (CNB-CSIC, Madrid). Moth cytochrome *c* (MCCp) 88–103 peptide (ANERADLIAYLKQATK) was purchased from GenScript. Other reagents used were mouse GM-CSF (Peprotech), LPS (Sigma-Aldrich), streptavidin microbeads (Miltenyi Biotec), streptavidin-PercP (Becton Dickinson), poly-L-lysine (Sigma-Aldrich), CellTrace Violet, Alexa Fluor568-phalloidin (both from Life Technologies), 7-AAD Viability Staining Solution (eBiosciences), and Live/Dead Fixable dead cell stain (Thermo Fisher). Percp-Streptavidin (1:300 dilution, BD Biosciences), Allophycocyanin-labelled dextramers specific for OVA H-2Kb (257-SIINFEKL-264) were purchased from Immudex.

**CD4[+] T-cell transphagocytosis.** CD4[+] T cells from OT-II mice transphagocy-tosed Listeria-OVA or Listeria-WT as described[19,61]. Briefly, bacteria-infected BM-DC from WT C57BL/6 mice, OVAp-II-loaded to improve transphagocytosis[19], were allowed to form conjugates with CD4[+] T cells. BM-DC cells from WT C57BL/6 mice as well as CD4[+] T cells from OT-II mice expressed MHC-I H-2K[b/k]. In some experiments, BM-DC and CD4[+] T cells were produced or isolated from AND transgenic mice expressing H-2K[b/k] or H-2K[k/k]. In these experiments, BM-DC were loaded with MCCp to improve transphagocytosis. After 3 h DC/T-cell conjugate formation, gentamicin (100 μg/ml) was added to cultures to eliminate extracellular bacteria. After 24 h (48 h in some cases), tpCD4[+] T cells were purified by cell sorting (FACS Synergy; iCyt). In some experiments, conjugates were allowed to form in the absence of antigen. BM-DC cells were separated from CD4[+] T cells with polycarbonate transwells (0.4 μm pore size; Costar) in some experiments (Supplementary Fig. 2A).

**Western blotting.** SDS–PAGE and western blotting were carried out using standard procedures. Bacteria samples (2 × 10[8]) were loaded in each lane. Extracellular medium was concentrated 50× using a 10 kDa Centricon (Millipore) before loading onto SDS-polyacrylamide gels. Polyclonal anti-OVA antibody was detected using HRP-coupled secondary antibodies and developed by chemiluminescence.

**CD8[+] T-cell proliferation assays.** Cell sorter-purified tpCD4[+] T cells were incubated with naïve OT-I mouse CD8[+] T cells, previously stained with CellTrace Violet to quantify proliferation by flow cytometry (FACSAria; BD). In every cell division, the proliferating population lost fluorescence, observed as a shift to the left in the histogram[24]; only live cells (negatively stained for 7AAD) were analyzed. As positive controls, OT-I CD8[+] T cells were incubated with OVAp-I-loaded BM-DC or with antibodies to CD3 (clone 17A2, eBioscience; 5 μg/ml, coated on plate) and CD28 (clone 37.51, BD Pharmingen, soluble 2 μg/ml).

To analyze CD8[+] T-cell proliferation induced by injected tpCD4[+] T cells in vivo, 5 × 10[6] naïve CD8[+] T cells (from CD45.1[+] OT-I mice), CellTrace Violet-stained, were injected i.v. into recipient mice (CD45.2[+] C57BL/6). After 24 h, tpCD4[+] T cells were adoptively transferred into mice. Three days after the second inoculation, spleens were isolated to measure CD8[+]CD45.1[+] T-cell proliferation by flow cytometry.

To test transphagocytosis and antigen presentation by CD4[+] T cells in the context of an in vivo bacterial infection, we performed a bone marrow transplant (Supplementary Fig. 3A). C57BL/6 mice were γ-irradiated (10 Gy) and transplanted with 3.8 × 10[6] bone marrow cells from H-2K[k] mice. After 30 days, CD4[+] T cells isolated by cell sorting from lymph nodes of AND mice (H-2K[b] or H-2K[k]) were adoptively transferred i.v. (4 × 10[6] cells per mouse), together with CD8[+] T cells from OT-I mice (4 × 10[6] cells per mouse) and MCCp (15 μg per mouse). These CD45.1[+]CD8[+] T cells were CellTrace Violet-stained before transfer. In addition, recipient mice were challenged i.v. with Listeria-OVA (10[4] bacteria per mouse). At 5 days post-infection, spleens were isolated and proliferation of the transferred OT-I cells was detected by flow cytometry (CellTrace Violet decay). To control of bone marrow elimination in recipient mice, the H-2K[b] allele was tested; only negative mice were analyzed.

**Analysis of memory CD8$^+$ T cells during Listeria infection**. Rag-1$^{-/-}$ recipient mice were divided into three experimental groups; mice were injected i.v. with (1) $5 \times 10^5$ naive CD8$^+$ T cells from WT C57BL/6 mice, (2) $5 \times 10^5$ naive CD8$^+$ T cells from WT C57BL/6 mice plus $5 \times 10^5$ CD4$^+$ T cells from Tap1$^{-/-}$ mice, or (3) $5 \times 10^5$ naive CD8$^+$ T cells plus $5 \times 10^5$ CD4$^+$ T cells, both from WT C57BL/6 mice. Naïve CD8$^+$ T cells were isolated from spleens by cell sorting. One day after cell transfer, Listeria-WT was injected i.v. ($10^3$ bacteria per mouse). Spleens were harvested after 5, 12 and 30 days to analyze CD8$^+$ T-cell phenotype by flow cytometry. Splenocytes were stained with antibodies to CD8, CD3, CD62L, CD44, CD127, PD-1 and CCR-7 conjugated with various fluorochromes. Mice received a secondary challenge of $5 \times 10^4$ Listeria-WT; after 2 days, spleen CFU were counted in agar plates.

Additionally, Rag-1$^{-/-}$ recipient mice were transferred i.v. with 1) $1 \times 10^6$ naïve CD8$^+$ T cells from CD45.1 WT C57BL/6 mice, plus $1 \times 10^6$ CD4$^+$ T cells from WT mice, or 2) $1 \times 10^6$ naïve CD8$^+$ T cells from CD45.1 WT C57BL/6 mice, plus $1 \times 10^6$ CD4$^+$ T cells from B2m$^{-/-}$ mice. One day after cell transfer, Listeria-OVA2 was injected i.v. ($10^3$ bacteria/mouse). Spleens were collected 30 days after challenge to analyze CD8$^+$ T-cell phenotype by flow cytometry. Splenocytes were stained with antibodies to CD8, CD45.1, CD62L, CD44, conjugated with various fluorochromes and with dextramers specific for OVA H-2K$^b$ (SIINFEKL).

**Immunofluorescence microscopy**. tpCD4$^+$ and naive CD8$^+$ T cells were allowed to form conjugates (1 h), then fixed with 4% paraformaldehyde in PBS. CD8$^+$ T cells were prestained with CellTrace Violet. Samples were permeabilized with 0.1% Triton X-100 in PBS before staining with indicated antibodies. F-actin was detected using fluorescently tagged phalloidin. Samples were visualized by confocal microscopy (Leica TCS-SP5; ×63 lens, controlled by Leica LAS AF). Images were analyzed with ImageJ software (NIH).

**Quantification the actin accumulation at the IS**. Cellular contacts between tpCD4$^+$ T cells and naïve CD8$^+$ T cells was prepared and visualized by confocal microscopy as described above. In order to quantify the amount of actin accumulated at the IS, confocal images were analysed using the software *Synapse measures*. *Synapse measures* can be used as a plugin for *Image J* and allowed us to accurately quantify the ratio between the immunofluorescence intensity of CD8$^+$ T-cell actin at the IS with that remained in the rest of the CD8$^+$ T cell (and taking into account the actin present in the tpCD4$^+$ T cells and the background signal). A detailed description of the *Synapse Measures* program, including the algorithms used, is described in ref. [31].

**Cytotoxicity assay**. Effector cytotoxic CD8$^+$ T cells were prepared from naive CD8$^+$ T cells from OT-I mice, activated with Listeria-OVA tpCD4$^+$ T cells (7 days). CD8$^+$ T cells from OT-I mice activated by OVAp-I-loaded splenocytes were used as positive controls.

EL-4 cells were incubated alone or with 0.5 μM OVAp-I (1 h). After washing with PBS, OVAp-I-loaded EL-4 cells were stained with 5 μM CellTrace Violet and unloaded EL-4 cells with 0.5 μM CellTrace Violet. After washing with RPMI 1640 (with FCS), the populations were mixed and incubated with CTL at various ratios (5:1, 2:1, 1:1, 0.5:1 EL-4:CTL; 4 h), then analyzed by flow cytometry. Specific cytotoxicity was calculated as $1-($% EL-4 CellViolet$^{high}$/% EL-4 CellViolet$^{low}) \times 100$[32]. Relative cytotoxicity was calculated by subtracting the specific cytotoxicity of the negative control (EL-4 cells incubated without CTL).

**Anti-tumour assay**. Listeria-WT CD4$^+$ T cells (negative control) or Listeria-OVA CD4$^+$ T cells were prepared as above. At 24 h post-transphagocytosis, tpCD4$^+$ T cells were reisolated by cell sorting and resuspended in PBS. Naïve CD8$^+$ T cells from OT-I mice were purified on magnetic columns as above and resuspended in PBS. B16-OVA cells ($5 \times 10^5$) were injected subcutaneously (s.c.) into the mid-right flank of C57BL/6 recipient mice. Mice were divided into three groups and adoptively transferred i.v. with PBS (group 1), $5 \times 10^5$ Listeria-WT tpCD4$^+$ T cells (group 2), or Listeria-OVA tpCD4$^+$ T cells (group 3). All groups were simultaneously adoptively transferred with $10^3$ OT-I CD8$^+$ T cells in a single i.v. injection. Tumour growth was measured every 2–3 days with a dial caliper, and areas determined by multiplying length by width. Tumour growth was flat, rendering tumour volume measurements unreliable. Experimental groups were assigned randomly and measurements made in a double-blind manner. Mice were killed when tumours reached 300 mm$^2$ in accordance with endpoints of the EU guidelines for experimental animals.

**Statistics**. To test whether treatment significantly reduced the odds ratio of developing a tumour, we used a one-sided Fisher's exact test to test the independence of tumour count and treatment given (treatment, Listeria-OVA tpCD4$^+$ T cells; controls, PBS or Listeria-WT tpCD4$^+$ T cells) as implemented in the R library exact $2 \times 2$ (Fig. 5b). The hypothesis that the odds ratio is the same in both conditions was rejected with a value $P = 0.004525$. For other experiments (Fig. 4c, g) we used one-way analysis of variance (ANOVA) and multiple mean comparisons corrected with the Bonferroni test. Data in Supplementary Fig. 1F were analyzed by two-way ANOVA and multiple mean comparisons with the Bonferroni correction. Data in Fig. 3f were analyzed by Mann–Whitney test. Data

in Supplementary Fig. 4F and G were analyzed by unpaired t test. Differences were considered significant at $p \leq 0.05$.

**Data availability**. The data that support the findings of this study are available from the corresponding author upon request.

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

## Acknowledgements

We are grateful to Laura Díaz-Muñoz for technical help with flow cytometry, Carlos Oscar Sánchez-Sorzano for help with statistics, Ignacio Melero for providing the B16-OVA melanoma cell line, Peter Klatt for critical reading of the manuscript and Catherine Mark for editorial assistance. J.M.G.-G. is supported by the Miguel Servet Program (Instituto de Salud Carlos III; ISCIII) and V.Z. by the ISCIII. This work was supported by grants from the Spanish Ministries of Science and Technology (MICINN; BFU2011-29450 to E.V.) and of Economy and Competitiveness (MINECO; SAF2014-56716-REDT and BFU2014-59585-R to E.V., SAF2014-55579-R to F.S.M., SAF2013-47975-R to B.A., SAF2014-58895-JIN to A.C.-A.), the ISCIII (PI14/00526; CP11/00145; CPII16/00022 to J.M.G.-G.), the Fundación Ramón Areces (to J.M.G.-G.), the Madrid regional government (INDISNET-S2011/BMD-2332 to F.S.M.) and the European Research Council (ERC-2011-AdG 294340-GENTRIS to F.S.M.; ERC 2013-AdG 334763 NOVARIPP to B.A.). F.S.M. and J.M.G.-G. are also financed by CIBER Cardiovascular, Spain.

## Author contributions

A.C.A. conducted and analyzed most of the experiments and helped both in the design of the experiments and in the writing of the manuscript. A.C.A., G.R.S., J.O.P., V.Z and M.T.T., conducted most of the experiments. A.M.R., V.B. and A.B. helped at different levels in the experiments involving the melanoma B-16OVA, G.H. provided support in some in vivo experiments. J.M. G.G., B.A. & F.S.M. critically read the manuscript and gave advice in the design of some experiments. E.V. designed the experiments, supervised and analysed the experiments, and wrote the manuscript.

## Additional information

**Competing interests:** The authors declare no competing financial interests.

