## [Peer review file · Nature Communications]

Reviewers' comments:

Reviewer #1 (Remarks to the Author):

This manuscript focuses on the role of CD4⁺ T cells that act as antigen presenting cells during Listeria infection, and explores their therapeutic potential. It is a follow-up to a recent paper by this group showing that prolonged CD4 interaction with Listeria-infected APCs results in Listeria infection of the CD4 cell. It is an intriguing concept, and the authors employ a variety of clever approaches to demonstrate that this process can convert CD4s into APCs that can activate CD4 T cells. They go on to suggest that in vivo activation of OT-1 via these CD4 APCs can enhance their ability to form memory upon subsequent Lm-Ova exposure. Lastly, they show that co-administration of OT-1 and CD4 APCs can enhance control of B16-OVA tumors. Overall, this is an intriguing observation that could change the way we think about T cell activation in vivo, and the experiments showing that CD4 APCs can activate naïve CD8 T cells in vitro and in vivo are well done and convincing. I have some concerns, however, about the relevance of this antigen presentation pathway during an actual infection, and the conclusion that CD4 APCs are uniquely suited to induce memory.

1. It is not clear the extent to which CD4 APCs are relevant in the context of a robust immune response to Listeria with many other activated APCs around. While the authors show in Fig. 4 that CD4 APCs that prime CD8 T cells prior to Listeria infection can promote better memory formation, there is no indication of the requirement for this pathway in the generation of CD8 effector and memory responses following infection of a normal mouse. A better experiment would be to assess effector and memory CTL formation in a mouse in which the ability of CD4 to present MHC Class I antigens is impaired during a normal Listeria infection, without exogenous transfer.
2. By the time CD4 T cells capture and present antigen during the context of a normal Listeria response, it is likely that CD8 T cells are already activated. Therefore, I question the role of CD4 APCs in priming naïve CD8 T cells, particularly in the formation of memory. Do CD4 APCs impact the formation of CD8 memory if they are instead transferred at the time of infection, or after the infection is initiated?
3. In Figure 4, CD4 APCs are transferred into Rag KO recipients a day prior to infection with Listeria. It is not clear if the enhanced memory seen in this scenario is due to a special property of the CD4 APCs themselves, or simply the impact of initiating the immunization one day prior to infection. For example, would a similar result be seen of mice were immunized by other means prior to Listeria infection, such as with antigen presenting DCs? Similarly, does immunization with CD4 APCs in the B16-OVA model have unique immunotherapeutic potential compared to DC immunization?
4. The authors repeatedly assert that CD4 APCs "cross-prime" naïve CD8 T cells (including in the title!). Given that this is a Listeria model, a pathogen that accesses the cytosol, they have not shown this, and there is no evidence of actual cross-priming. It seems more likely that CD4s process and present MHC I antigens conventionally after Listeria accesses the cytosol. The manuscript should be amended to remove this claim, or the authors should show evidence of actual cross-priming.
5. It might be helpful to include a more extensive discussion on the potential role of CD4 APCs in other types of infections. Do the authors predict that this is a phenomenon unique to Listeria?
6. There are numerous grammatical and writing errors throughout, and the authors should consider obtaining editing assistance to enhance clarity.

Reviewer #2 (Remarks to the Author):

In this manuscript, Cruz-Adalia et al demonstrate that CD4 T cells are able to cross-present antigens they obtained via transphagocytosis from DCs. In order to do this, they incubated CD4 T cells with DCs that were previously infected with Lm-OVA or Lm and loaded with the peptide epitope of the CD4 T cells. Then, they sorter the CD4 T cells from the coculture and incubated them with CD8 T cells.

They demonstrated that DC infection with Lm-OVA resulted in CD8 T cells proliferation and upregulation of activation markers. Transwell experiments demonstrated that a direct interaction between the CD4 T cell and the DC were required, pointing out the possibility that DCs pass on antigens to the CD4 T cells. Experiments using DCs from H2Kk mice excluded that peptide-loaded MHC I molecules were transferred from the DC to the T cell, pointing out the necessity of antigen processing within the CD4 T cell, which was further supported by using T cells from TAP^{-/-} mice. Subsequently, they demonstrated similar mechanisms in vivo. Injected CD8 T cells proliferated in the presence of matching tiCD4 T cells and a memory CD8 T cell response was induced. Finally, they demonstrated in a tumor model that vaccination with tiCD4 T cells can induce antigen-specific CD8 T cell anti-tumor responses.

The authors provide a whole new concept in putative CD8 T cell activation. Although I am really skeptical regarding the physiological relevance compared to "classical" DC-induced CD8 T cell activation, the authors convincingly show that antigens can be transported from the DC to the CD4 T cells, these cells can process them and loaded antigen-derived peptides on MHC I and subsequently activate antigen-specific T cells.

Major remarks:

1. The authors demonstrated that transphagocytosed antigens can be cross-presented to T cells. However, it is important that the relevance of this process compared to DC-mediated CD8 T cell activation is investigated more carefully. Therefore, the authors should compare in vivo experiments the efficiency of CD8 T cell activation by tiCD4 T cells versus DCs
2. All experiments were performed using tiCD4 T cells primed by infected DCs in the presence of OVA-peptide, in order to increase transphagocytosis. Does this mean that cross-presentation of OVA by DCs is not sufficient to efficiently generate tiCD4 T cells? The authors should provide further evidence on CD8 T cell activation in the absence of OVA peptide
3. It is very hard to believe that CD8 T cells primed by tiCD4 T cells proliferate equally well compared to T cells activated by infected DCs themselves and even by peptide-loaded DCs or after CD3/CD28 stimulation. However, CD3/CD28 is a very potent T cell activator. It is very surprising that transphagocytosis and cross-priming by CD4 T cells are equally efficient compared to direct activation by peptide-loaded DCs, which express significantly more co-stimulatory molecules and are superior in secretion of pro-inflammatory cytokines. Similarly, cytotoxic T cell activity after priming with tiCD4 T cells is superior compared to peptide-loaded splenocytes, which also include professional APCs. The authors should provide an explanation what could cause such superior T cell activating potential

Other remarks:

- To me, it is unclear how the authors conclude from figure 2A that antigen processing needs to occur in the T cells (p7 - line 150). This should be explained better
- Actin organization in Figure 3D and 3E should be quantified

Reviewers' comments:

Reviewer #1 (Remarks to the Author):

This manuscript focuses on the role of CD4+ T cells that act as antigen presenting cells during Listeria infection, and explores their therapeutic potential. It is a follow-up to a recent paper by this group showing that prolonged CD4 interaction with Listeria-infected APCs results in Listeria infection of the CD4 cell. It is an intriguing concept, and the authors employ a variety of clever approaches to demonstrate that this process can convert CD4s into APCs that can activate CD4 T cells. They go on to suggest that in vivo activation of OT-1 via these CD4 APCs can enhance their ability to form memory upon subsequent Lm-Ova exposure. Lastly, they show that co-administration of OT-1 and CD4 APCs can enhance control of B16-OVA tumors. Overall, this is an intriguing observation that could change the way we think about T cell activation in vivo, and the experiments showing that CD4 APCs can activate naïve CD8 T cells in vitro and in vivo are well done and convincing. I have some

concerns, however, about the relevance of this antigen presentation pathway during an actual infection, and the conclusion that CD4 APCs are uniquely suited to induce memory.

-AU. We did not exclude the (well-known) role of DC in T cell memory formation. The data we presented here, expand the knowledge of the roles of CD4⁺ T cells during CD8⁺ T cell memory formation, though to be limited to license DC. Now, we show that tiCD4⁺ T cells, in addition, are also able to directly prime naïve CD8⁺ T cells inducing central memory. In this regard, we also show that the fate of memory CD8⁺ T cell is different depending if their activation is mediated by DC or CD4⁺ T cells

1. It is not clear the extent to which CD4 APCs are relevant in the context of a robust immune response to *Listeria* with many other activated APCs around.

-AU. We used Rag-1^{-/-} mice as recipients in order to analyze the CD4⁺ T cell dependent antigen presentation in the context of a robust APC response. We have already shown the tiCD4⁺ T cell-dependent activation of naïve CD8⁺ T cells in a context of impaired DC presentation using H2-K^k reconstituted mice (Supp. Fig. 3)

Rag-1^{-/-} mice which lack B and T lymphocytes, have an intact, robust, innate immune response. Even in this context, tiCD4⁺ T cell-mediated antigen presentation resulted fundamental to develop full CD8⁺ T cell central memory. We have compared the generation of central memory T cells in the presence of WT tiCD4⁺ T cells, Tap1^{-/-} tiCD4⁺ T cells (with limited antigen presentation capacity) or in the absence of CD4⁺ T cells. Central memory generation was reduced in the absence of CD4⁺ T cells or in the presence of Tap1^{-/-} tiCD4⁺ T cells

Now, to strengthen the previous results, we include a new experiment using also Rag-1^{-/-} mice as recipients, to which we transferred WT or $\beta 2m^{-/-}$ (with low levels of MHC-I molecules) CD4⁺ T cells together with WT CD8⁺ T cells. Transferred mice were later infected with *Listeria-OVA2* (*Listeria monocytogenes* expressing OVA generated in our laboratory; Supplementary Figure 2 C-D). This experiment is presented in Supp. Fig 4 E-G and confirms that the generation of CD8⁺ T central memory was significantly reduced in the presence of CD4⁺ T cells with limited antigen presentation abilities. In addition, the expansion of CD8⁺ T cells responding to OVA was decreased when used $\beta 2m^{-/-}$ CD4⁺ T cells.

These data highlight the role *in vivo* of the tiCD4⁺ T cell-mediated antigen presentation in CD8⁺ T cell activation and in the generation of central memory in the context of a complete innate immune response.

While the authors show in Fig. 4 that CD4 APCs that prime CD8 T cells prior to *Listeria* infection can promote better memory formation, there is no indication of the requirement for this pathway in the generation of CD8 effector and memory responses following infection of a normal mouse.

A better experiment would be to assess effector and memory CTL formation in a mouse in which the ability of CD4 to present MHC Class I antigens is impaired during a normal *Listeria* infection, without exogenous transfer.

-AU. We probably did not explain our experimental data in a proper way. The CD4⁺ T cells in Fig 4 were not transphagocytic at the time of cell transfer (we transferred CD4⁺ T cells directly after isolation from lymph nodes) and therefore they cannot prime CD8⁺ T cells in the absence of *Listeria* (CD4⁺ T cells are able to present antigens only after bacterial capture, Fig. 1). Therefore, the APC ability of the tiCD4⁺ T cells only occurs after bacterial infection, *in vivo*, never prior to bacterial infection.

-AU. The experiment suggested by the referee needs the generation of mice with CD4⁺ T cells impaired (or reduced) in antigen presentation. For example, Tap1^{-/-}, or β2m^{-/-}, specifically in the CD4 lineage, which will take a very long time, almost one year, to be set up. The experiments of antigen presentation using these CD4 lineage specific KO mice would be very interesting to further dissect the fine molecular mechanism used by tiCD4⁺ T cells for antigen presentation, that I think it is an excessive demand and it is out of the scope of our present study.

On the other hand, at this point we think these experiments would not provide a substantial difference compared to those that we have already shown, including the new data using β2m^{-/-} CD4⁺ T cells detailed above. In our opinion we clearly show that tiCD4⁺ T cells are able to present antigens, which represent a breakthrough in immunology. Dissecting the precise and detailed pathways using by CD4⁺ cells to become APCs will take several years of research.

2. By the time CD4 T cells capture and present antigen during the context of a normal *Listeria* response, it is likely that CD8 T cell are already activated. Therefore, I question the role of CD4 APCs in priming naïve CD8 T cells, particularly in the formation of memory. Do CD4 APCs impact the formation of CD8 memory if they are instead transferred at the time of infection, or after the infection is initiated?

-AU. The CD4⁺ T cells in Fig 4 were not transphagocytic at the time of cell transfer and therefore they cannot prime CD8⁺ T cells in the absence of *Listeria*.

-AU. In spite of this, we performed the experiment suggested by the referee and compare the generation of memory CD8⁺ T cell (CD44⁺ CD62L^{high}) in Rag-1^{-/-} recipient mice transferred with WT CD4⁺ and WT CD8⁺ T cells and infected with *Listeria monocytogenes* before or after cellular transfer. No significant differences were observed (Fig 1 for referee) indicating that CD8⁺ T cells were not preactivated before bacterial challenge.

3. In Figure 4, CD4 APCs are transferred into Rag KO recipients a day prior to infection with *Listeria*. It is not clear if the enhanced memory seen in this scenario is due to a special property of the CD4 APCs themselves, or simply the impact of initiating the immunization one day prior to infection.

For example, would a similar result be seen of mice were immunized by other means prior to *Listeria* infection, such as with antigen presenting DCs? Similarly, does immunization with CD4 APCs in the B16-OVA model have unique immunotherapeutic potential compared to DC immunization?

-AU. The generation of memory CD8⁺ T cell (CD44⁺ CD62L^{high}) was not affected by infecting with *Listeria* before or after CD4⁺ T cell transfer (Fig 1 for referee)

-AU. As suggested by the referee, we compared the generation of memory CD8⁺ T cell in Rag-1^{-/-} mice transferred with CD8⁺ T cells together with CD4⁺ T cells, BM-DC or no APC. The day after, recipient mice were challenged with *Listeria* and 30 days later CD8⁺ T cells were analyzed (Fig 2 for referees). The % of central memory CD8⁺ T cells was significant reduced in the absence of CD4⁺ T cells.

-AU. The effect of DC- vs tiCD4⁺ T cell-based antigen presentation on CD8⁺ T cell central memory generation was also shown in Fig 4, and in new Supp. Fig. 4 E-G. Rag-1^{-/-} mice have a complete and robust innate immune response (including DC-based antigen presentation) but the maximal generation of central memory was observed only in the presence of WT CD4⁺ T cells, fully capable of presenting antigens. If the CD4⁺ T cells are defective in antigen presentation (Tap-1^{-/-} or β 2m^{-/-}) or in the absence of CD4⁺ T cells, the generation of central memory CD8⁺ T cells (by endogenous DC) was significantly reduced. Note that CD4⁺ T cells defective in antigen presentation expressed the same amount of CD40L than WT (Supp. Fig. 4C).

4. The authors repeatedly assert that CD4 APCs “cross-prime” naïve CD8 T cells (including in the title!). Given that this is a *Listeria* model, a pathogen that accesses the cytosol, they have not shown this, and there is no evidence of actual cross-priming. It seems more likely that CD4s process and present MHC I antigens conventionally after *Listeria* accesses the cytosol. The manuscript should be amended to remove this claim, or the authors should show evidence of actual cross-priming.

-AU. Taken into account that (1) OVA antigen was produced by bacteria (exogenous generation of antigen); (2) *Listeria* degradation inside tiCD4⁺ T cells occurs in a lysosomal compartment¹; and (3) Antigen presentation occurs (at least partially) through the TAP1 pathway, the canonical cross presentation pathway in DC (Fig. 2-G), we can conclude that tiCD4⁺ T cell-mediated activation of CD8⁺ T cells occurs though cross-presentation. It is in our aim to finely define in the future the exact route for cross-presenting antigens using by tiCD4⁺ T cells and compare this cross-presentation with that observed in DC.

5. It might be helpful to include a more extensive discussion on the potential role of CD4 APCs in other types of infections. Do the authors predict that this is a phenomenon unique to *Listeria*?

-AU. We have shown that different bacteria can be captured and degraded by CD4⁺T cells¹, and we are in the way to determine whether other bacteria could improve the APC abilities of tiCD4⁺ T cells. We have included a paragraph in the expanded the discussion.

6. There are numerous grammatical and writing errors throughout, and the authors should consider obtaining editing assistance to enhance clarity.

-AU. The new version has been edited by an English-speaking native professional editor specialized in scientific texts.

Reviewer #2 (Remarks to the Author):

In this manuscript, Cruz-Adalia et al demonstrate that CD4 T cells are able to cross-present antigens they obtained via transphagocytosis from DCs. In order to do this, they incubated CD4 T cells with DCs that were previously infected with Lm-OVA or Lm and loaded with the peptide epitope of the CD4 T cells. Then, they sorter the CD4 T cells from the coculture and incubated them with CD8 T cells. They demonstrated that DC infection with Lm-OVA resulted in CD8 T cells proliferation and upregulation of activation markers. Transwell experiments demonstrated that a direct interaction between the CD4 T cell and the DC were required, pointing out the possibility that DCs pass on antigens to the CD4 T cells. Experiments using DCs from H2Kk mice excluded that peptide-loaded MHC I molecules were transferred from the DC to the T cell, pointing out the necessity of antigen processing within the CD4 T cell, which was further supported by using T cells from TAP^{-/-} mice. Subsequently, they demonstrated similar mechanisms in vivo. Injected CT8 T cells proliferated in the presence of matching tiCD4 T cells and a memory CD8 T cell response was induced. Finally, they demonstrated in a tumor model that vaccination with tiCD4 T cells can induce antigen-specific CD8 T cell anti-tumor responses. The authors provide a whole new concept in putative CD8 T cell activation. Although I am really skeptical regarding the physiological relevance compared to "classical" DC-induced CD8 T cell activation, the authors convincingly show that antigens can be transported from the DC to the CD4 T cells, these cells can process them and loaded antigen-derived peptides on MHC I and subsequently activate antigen-specific T cells.

Major remarks: 1. The authors demonstrated that transphagocytosed antigens can be cross-presented to T cells. However, it is important that the relevance of this process compared to DC-mediated CD8 T cell activation is investigated more carefully. Therefore, the authors should compare in in vivo experiments the efficiency of CD8 T cell activation by tiCD4 T cells versus DCs

-AU. As suggested by the referee, we compared the generation of memory CD8⁺ T cell in Rag-1^{-/-} mice transferred with CD8⁺T cells together with CD4⁺ T cells, BM-DC or no APC. The day after, recipient mice were challenged with *Listeria* and 30 days later CD8⁺ T cells were analyzed (Fig 2 for referees). The % of central memory CD8⁺ T cells was significantly reduced in the absence of CD4⁺ T cells.

-AU. Note that the effect of DC- vs tiCD4⁺ T cell-based antigen presentation on CD8⁺ T cell central memory generation was also shown in Fig 4 and in addition in new Supp. Fig. 4 E-G. Rag-1^{-/-} mice have a complete and robust innate immune response (including DC-based antigen presentation) but the maximal generation of central memory was observed only in the presence of WT CD4⁺ T cells, fully capable of presenting antigens. If the CD4⁺ T cells are defective in antigen presentation (Tap-1^{-/-} or β2m^{-/-}) or in the absence of CD4⁺ T cells, the generation of central memory CD8⁺ T cells (by endogenous DC) was significantly reduced.

2. All experiments were performed using tiCD4 T cells primed by infected DCs in the presence of OVA-peptide, in order to increase transphagocytosis. Does this mean that cross-presentation of OVA by DCs is not sufficient to efficiently generate tiCD4 T cells? The authors should provide further evidence on CD8 T cell activation in the absence of OVA peptide

-AU. Antigen presentation enhances, but is not necessary for transphagocytosis ¹.

-AU. There are several experiments performed in the absence of OVA peptide (or any other soluble antigen). For example, the *in vitro* experiments using TAP1^{-/-} mice (Fig 2G). CD4⁺ T cells obtained from WT (B6), or TAP1^{-/-} mice, express the whole repertoire of TCR (they are not TCR transgenic, like OT-II mice), and the transphagocytosis was performed in the absence of OVA peptide. The same applies for the *in vivo* experiments showed in Fig. 4. No OVA peptide was provided to mice.

- AU. In the new Supplementary Figure 2D, naïve OT-I CD8⁺ T cell activation was induced by CD4⁺ T cells isolated from WT (C57BL/6) mice that transphagocytosed *Listeria*-OVA2 from BM-DCs without OVAp (nor any other antigen)

-AU. In the *in vitro* experiments shown in Fig 2B-C (H2-K^k, H2-K^b experiments) transphagocytosis was performed in the presence of soluble antigen MCCp, not OVAp.

3. It is very hard to believe that CD8 T cells primed by tiCD4 T cells proliferate equally well compared to T cells activated by infected DCs themselves and even by peptide-loaded DCs or after CD3/CD28 stimulation. However, CD3/CD28 is a very potent T cell activator. It is very surprising that transphagocytosis and cross-priming by CD4 T cells are equally efficient compared to direct activation by peptide-loaded DCs, which express significantly more co-stimulatory molecules and are superior in secretion of pro-inflammatory cytokines.

-AU. We agree with the reviewer that these data are surprising, but they are very consistent over more than 10 independent experiments.

Similarly, cytotoxic T cell activity after priming with tiCD4^+ T cells is superior compared to peptide-loaded splenocytes, which also include professional APCs. The authors should provide an explanation what could cause such superior T cell activating potential

-AU. The experiment shows just the contrary. The cytotoxic activity of activated OT-I CD8 T cells resulted more potent when the APCs were splenocytes loaded with OVAp-I than tiCD4^+ T cells capturing *Listeria* OVA (Fig 3F). This experiment show that OT-I CD8⁺ T cells activated with tiCD4^+ T cells are cytotoxic, but they present less cytotoxic activity than that induced by OVAp-I-loaded splenocytes. Note that the amount of OVA antigen presented in cells loaded with OVAp-I is larger than cells expressing OVA antigen from captured/degraded *Listeria*-OVA

Other remarks:

- To me, it is unclear how the authors conclude from figure 2A that antigen processing needs to occur in the T cells (p7 - line 150). This should be explained better

-AU. If the tiCD4^+ activation of CD8⁺ T cells was due to MHC-I/antigen capture from DC (decorated with OVAp-I) by trogocytosis, CD8⁺ T cells would be activated similarly in both conditions (*Listeria* WT and *Listeria*-OVA). We nonetheless found that *Listeria*-OVA tiCD4^+ T cells but not *Listeria*-WT tiCD4^+ T cells caused strong activation of CD8⁺ T cells. These results were confirmed by the experiments shown in Fig 2B-G using DCs expressing H2-K^k. We have rewritten the text in order to better convey the message.

- Actin organization in Figure 3D and 3E should be quantified

-AU. We have quantified actin accumulation at the immunological synapse (new Fig. 3F) using the software that we developed (a plugin for ImageJ; "Synapse Measures") to quantify protein accumulation at the immunological synapses²).

Fig 1 (for referees)

*Fig 1. Phenotypic analysis of memory CD8⁺ T cell generation in vivo. Rag1^{-/-} mice were challenged with *L. monocytogenes* (10³ bacteria/mice; i.v.) and adoptively transferred i.v. with WT CD4⁺ T plus WT CD8⁺ T cells the day before (bi; group 1) or the day after (ai; group 2) infection. 30 days after challenge, the CD8⁺ T cell population from spleen was analyzed. The percentage of central memory CD8⁺ T cells (CD44⁺CD62L^{high}) is shown. Data shown as mean ± SD for 3 mice (group 1, black) and 2 mice (group 2, grey). No significant differences were observed*

Fig 2 (for referees)

*Fig 2. Phenotypic analysis of memory CD8⁺ T cell generation in vivo. Rag1^{-/-} mice were adoptively transferred i.v. with 1) only WT CD8⁺ T cells, 2) WT CD8⁺ T cells plus BMDC, and 3) WT CD8⁺ T plus WT CD4⁺ T cells. Recipient mice were challenged the day after cellular transfer with *L. monocytogenes* (10³ bacteria/mice; i.v.). 30 days after challenge, the CD8⁺ T cell population from spleen was analyzed. The percentage of central memory CD8⁺ T cells (CD44⁺CD62L^{high}) is shown. Data shown as mean ± SD for 3 mice/group. * p<0.05, ** p<0.005 analyzed by ANOVA and Bonferroni post-test.*

1. Cruz-Adalia, A. *et al.* T Cells Kill Bacteria Captured by Transinfection from Dendritic Cells and Confer Protection in Mice. *Cell Host Microbe* **15**, 611–622 (2014).
2. Calabia-Linares, C. *et al.* Endosomal clathrin drives actin accumulation at the immunological synapse. *J. Cell Sci.* **124**, 820–830 (2011).

REVIEWERS' COMMENTS:

Reviewer #1 (Remarks to the Author):

The authors provided several new experiments and expanded discussion to address previous concerns. Overall the findings are new and novel and the changes are satisfactory.

I continue to be confused by the authors' insistence on the term cross-presentation to describe the results, particularly considering that this finding is not central to their overall conclusions. The authors claim that degradation occurs only in the endosome in CD4 T cells, and therefore antigen presentation must occur through cross-presentation. Escape from the endosome into the cytosol is a normal phase of the *Listeria* life cycle, and seems like a much more likely means of access to MHC Class I presentation pathways. Do the authors have evidence that *Listeria* does not access the cytosol in CD4 T cells? The fact that *Listeria* secretes OVA within the endosome, or that the process is TAP-dependent, is irrelevant in distinguishing cross-presentation from conventional antigen presentation for this pathogen. The authors should show evidence of cross-presentation or amend their conclusions.

Reviewer #2 (Remarks to the Author):

Although I am still surprised by their results and sceptical about the physiological relevance of CD8 T cell activation by CD4 T cells, the authors seem to provide compelling evidence for their novel concept. It will be interesting to see in future studies in which conditions this process becomes relevant.

Please, find enclosed our response to the reviewer's comments.

REVIEWERS' COMMENTS:

Reviewer #1 (Remarks to the Author):

The authors provided several new experiments and expanded discussion to address previous concerns. Overall the finding new and novel and the changes are satisfactory.

I continue to be confused by the authors insistence on the term cross-presentation to describe the results, particularly considering that this finding is not central to their overall conclusions. The authors claim that degradation occurs only in the endosome in CD4 T cells, and therefore antigen presentation must occur through cross-presentation. Escape from the endosome into the cytosol is a normal phase of the Listeria life cycle, and seems like a much more likely means of access to MHC Class I presentation pathways. Do the authors have evidence that Listeria does not access the cytosol in CD4 T cells? The fact that Listeria secretes OVA within the endosome, or that the process is TAP-dependent, is irrelevant in distinguishing cross-presentation from conventional antigen presentation for this pathogen. The authors should show evidence of cross-presentation or amend their conclusions.

AU- • We systematically modified the manuscript to remove the sentences on crosspresentation by tpCD4+ T cells. We discuss about it in the discussion section showing the data that suggest that it might be crosspresentation and also the data indicated by referee 1 that goes in the opposite direction.

Reviewer #2 (Remarks to the Author):

Although I am still surprised by their results and sceptical about the physiological relevance of CD8 T cell activation by CD4 T cells, the authors seem to provide compelling evidence for their novel concept. It will be interesting to see in future studies in which conditions this process becomes relevant.

AU. We thank the positive comments of the referee. We are working in unveil the physiological relevance of tpCD4+ T cells antigen presentation in antitumour, anti bacterial, and anti viral fight.